# Development of on-farm AMF inoculum production for sustainable agriculture in Senegal

Christian Valentin Nadieline[1], Antoine le Quéré[2], Cheikh Ndiaye[3], Amadou Abib Diène[4], Francis Do Rego[5], Oumar Sadio [6], Yoro Idrissa Thioye[7], Marc Neyra[2], Cheikh Mouhamed Fadel Kébé[8], Tatiana Krasova Wade [5]*

1 Institut de Recherche pour le Développement (IRD), Laboratoire Commun de Microbiologie IRD/ISRA/UCAD (LCM), Centre de Recherche de Bel-Air, Dakar, Senegal, 2 Institut de Recherche pour le Développement (IRD), UMR Eco&Sols IRD/CIRAD/INRAE/Montpellier SupAgro, Montpellier, France, 3 Département de Biologie Végétale, Faculté des Sciences et Techniques, Université Cheikh Anta Diop de Dakar, Dakar-Fann, Senegal, 4 GIE Ndiénné, Commune de Darou Mousty, Village Darou Mousty, Senegal, 5 Institut de Recherche pour le Développement (IRD), UMR Eco&Sols IRD/CIRAD/INRAE/Montpellier SupAgro / IESOL, Centre de Recherche de Bel-Air, Dakar, Senegal, 6 Institut de Recherche pour le Développement (IRD), UMR 195 LEMAR, ISRA-CRODT /Pôle de Recherche de Hann, Dakar, Senegal, 7 Conseil National de Concertation et de Coopération des Ruraux (CNCR), Dakar, Senegal, 8 Ecole Supérieure Polytechnique de Dakar Département Génie Civil, Université Cheikh Anta Diop de Dakar, Dakar-Fann, Senegal

* tania.wade@ird.fr

**Data Availability Statement:** All relevant data are within the manuscript and its Supporting information files.

## Abstract

The integration of endomycorrhizal fungi into agricultural practices as inoculum offers the potential to improve plant productivity while reducing reliance on expensive chemical fertilizers, which are not only economically costly but also detrimental to the environment. Mycorrhizal fungi play a crucial role in facilitating plant access to essential mineral elements (such as Phosphorus, Potassium, etc.) and water, particularly in soils characterized by arid and semi-arid conditions. where these resources are often limited. Despite the obvious advantages, the development of arbuscular mycorrhizal fungi (AMF) inoculum production in Africa is progressing on a small scale. This research aims to address this limitation by exploring the feasibility of producing mycorrhizal inoculum on a semi-industrial farm scale, achieved through the control and stabilization of production parameters. Crop residues as peanut shell, rice husk, sugar cane bagasse and millet ears were tested in Leonard jars and pots as alternatives to conventional sand production substrate for the multiplication of mycorrhizal fungi *Glomus aggregatum* IR27, *Funneliformis mosseae*, *Rhizophagus irregulares* and *Glomus fasciculatum* R10. Significant results were obtained on the peanut shell. Under mass production conditions in farm scale, *Glomus aggregatum* IR27 showed the best mycorization characteristics with 19.76% intensity and 88.93% frequencies. The study highlighted the critical considerations of irrigation water salt content and substrate sterilization as essential parameters to ensure optimal development of mycorrhizal propagules. Water containing 0.5% salt inhibited the mycorrhization. This negative effect of salt was much more accentuated in unsterilized peanut shell substrate than in sterilized one. This experimental study constitutes a pioneering initiative, potentially replicable in other agricultural areas. Its

**Funding:** OAPI funding 27714 to T.K.W. and IRD funding Coup de pouce 2017 "MySen" to T.K.W.

**Competing interests:** The authors have declared that no competing interests exist.

sustainability is based on the simplicity and efficiency of the technology, which opens the prospect of increasing the number of AMF inoculum production units on a national scale in Senegal.

## Introduction

Agricultural activity within the West African sub-region is not sufficient to fulfill its raising population needs which also faces environmental changes. To address this challenges, commonly recommended solutions involve practices such as the application of chemical fertilizers, pesticide use, adoption of new seed varieties, and water control collectively referred to as agricultural intensification. However, these practices, while offering short-term gains, often result in long or medium-term degradation of resources and the environment, particularly contributing to groundwater pollution [1, 2].

Simultaneously, research outcomes have proposed various alternative strategies aimed at restoring soil fertility levels and increasing plant production in developing countries in Africa [1]. Among these strategies is the utilization of symbiotic microorganisms as inoculants, particularly arbuscular mycorrhizal fungi (AMF) in the form of biostimulant inoculum [3, 4].

Incorporating mycorrhizal fungi into agricultural practices represents a viable technical approach to enhance plant productivity while mitigating the need for costly and environmentally harmful chemical fertilizers. Through symbiosis, mycorrhizal fungi play a pivotal role in aiding plants to access mineral elements such as phosphorus and potassium, sodium, as well as water resources frequently limited in arid and semi-arid soils [1, 5–9], and even in improving of the essential oil profils [4] and fatty acids [10]. This inoculum serves as a biological stimulant; it is cost-effective, environmentally friendly, and easy to use.

A meta-analysis encompassing 290 greenhouse and field trials aimed at investigating the impact of various agricultural practices on mycorrhizal colonization revealed that inoculation led to a substantial increase in mycorrhizal colonization, reaching up to 29% [2, 11]. Notably, in Senegal, research findings dating back to 1994 demonstrated the effectiveness of arbuscular mycorrhizal fungi (AMF) in root colonization, extending to depths of up to 40 meters in association with *Acacia albida* in Sahelian zones [12]. Furthermore, mycorrhizal inoculation exhibited cost reduction and increased yields in cash crops [13], forestry [14, 15], various food crops [16–18]. Additionally, it was demonstrated across diverse agro-ecological zones adaptability to saline and water stress conditions [19] and improving of plant production in arid and semi-arid zones [20–23].

While certain countries have witnessed significant progress in AMF production and utilization, Sub-Saharan African farmers face challenges due to limited production and awareness of the associated benefits [1]. AMF inoculum production units are scarce in the region, with knowledge limited to Kenya and South Africa as of 2016 [1]. The production scale of mycorrhizal fungi inoculum in Africa remains modest due to technological constraints, underscoring the pivotal role of production technology and inoculum formulation support for sustainable application [24, 25].

In Senegal, mycorrhizal inoculation has gained social acceptance among farmer organizations, agricultural advisory services, and management structures, creating a demand for its widespread availability [26, 27]. Recognizing this demand, the current challenge is the mass production of high-quality inoculum to meet growing needs, as research laboratories have historically produced inoculum on a small scale for research purposes.

This study endeavors to explore the feasibility of producing elite AMF inoculum on a semi-industrial scale within agricultural production sites utilizing locally available means and

materials, with a focus on controlling and stabilizing production parameters. The objective is to establish a practical and sustainable approach that aligns with the local context. Ultimately, it contributes to stimulate the integration of symbiotic microorganisms into agricultural practices in the form of bio-stimulants.

## Material and methods

### Experimental sites

The experiments were conducted at the Joint Microbiology Laboratory IRD-ISRA-UCAD (LCM) located at the Bel Air Research Center in Dakar, Senegal, and in a greenhouse on a pilot site for mycorrhizal fungi inoculum production in the commune of Darou Mousty (Peanut Basin zone) in Senegal.

### Arbuscular mycorrhizal fungi (AMF) material

The initial mycorrhizal strains used for multiplication in the laboratory and on the pilot site (starters) were provided by the LCM. These strains included *Glomus aggregatum* IR27 [28], *Funneliformis mosseae* [29], *Rhizophagus irregularis* [30], and *Glomus fasciculatum* R10 [29]. The material consisted of a mixture of sand (as a culture substrate), fragments of mycorrhizal roots, spores, and AMF hyphae. The mycorrhization parameters of the initial strains, multiplied prior to multiplication on the pilot site, are presented in Table 1.

### AMF multiplication using agricultural residues as inoculum support in the lab

To explore the feasibility of substituting sand with agricultural residues from crop production, experiments were conducted in Leonard jars and pots in the laboratory. Various agricultural residues, such as peanut hulls, rice husks, millet ears, and sugarcane bagasse, were utilized either independently or in combination with sand (50/50, v/v). In Leonard jars, contained 50 ml of substrate, each treatment was repeated six times and in pots, contained 2 l of substrate, four times.

Prior to experimentation, these residues were crushed, humidified, and sterilized in an oven at 65°C for three days. The sand was autoclaved at 120°C for 2 hours. Corn (*Zea mays* L.) was chosen as the host plant that is justified by the properties of high mycotrophy and root biomass comprising a large number of fine roots, thus facilitating the establishment of mycorrhizae.

For multiplication, five grams (5 g) of a mixture containing the four AMF strains in equal proportions were used. The plants were irrigated with demineralized water. After three months of cultivation, watering was stopped for three weeks until the aerial part has completely dried out to induce stress in the plants and facilitate sporulation. The aerial parts of the corn plants were initially harvested to free the roots, and the inoculum was subsequently

**Table 1. Mycorrhization parameters (mean data) of AMF starters used for inoculum production on the pilot site.**

| AMF Strain | Mycorrhization frequency (%) | Mycorrhization intensity (%) | Number of spores ($100^{-1}$g of substrate) |
|---|---|---|---|
| *Glomus agregatum* IR27 | 100 | 92 | 1313 |
| *Funneliformis mosseae* | 95 | 90 | 1024 |
| *Glomus fasciculatum* | 90 | 85 | 917 |
| *Rhizophagus irregularis* | 100 | 95 | 1230 |

collected from each container, by blending the cut corn roots with the production support between all of the replicates to compose one random sample.

## Inoculum quality control

Quality control of amplified AMF involved assessing two parameters: mycorrhization intensity and frequency rate through corn root coloring and spore count by extraction and enumeration in the culture substrate.

In each sample, a portion of the corn roots was selected for coloring following the method of Philips and Hayman [31]. Briefly, the sampled roots were thoroughly rinsed in tap water to remove soil particles and then immersed in a 10% (w/v) KOH solution. KOH aided in discoloring the roots and emptying the cytoplasmic contents of the root cells. The tubes containing the roots and KOH were boiled in a water bath at 90°C for 30 minutes. Subsequently, the roots were thoroughly rinsed to eliminate KOH and immersed in a 0.05% (w/v) Trypan blue solution for staining in a water bath at 90°C for 30 minutes. Following this step, the roots were washed with water and stored in a cool place for histological observation. For each sample, approximately 1 cm-long root fragments were interposed between slides and lamellae, subsequently crushed in 90% glycerol, and subjected to microscopic observation at x 20 magnification. A total of 100 root fragments were taken randomly for each treatment. On slide contained 20 fragments, with 5 repetitions for each treatment. The identification of hyphae, vesicles, or arbuscules in the root enables the estimation of the colonization rate in the root sample. Arbuscular mycorrhizal fungi root colonization rate was assessed using the method outlined by Trouvelot et al. [32].

The intensity of root mycorrhization was calculated as:

$$I\% = (95n5 + 70n4 + 30n3 + 5n2 + n1)/\text{Total number of observed fragments}$$

where n5, n4, n3, n2, and n1 represent the number of fragments scored as 5, 4, 3, 2, and 1, respectively.

The frequency of root mycorhization was calculated as:

$$F\% = (\text{number of mycorrhizal fragments}/\text{total number of fragments observed}) \times 100$$

Spore extraction and quantification were conducted following the protocol established by Gerdemann and Nicolson [33]. Initially, 100 g of inoculum was combined with 1 liter of tap water, followed by stirring for 1 minute and settling for 30 seconds. The supernatant underwent filtration through a series of stacked sieves with diminishing mesh diameters (400 μm, 200 μm, 100 μm, 50 μm). Spores retained by the 200 μm and 50 μm sieves were gathered in distilled water and transferred to centrifugation tubes. Subsequently, two sucrose solutions (20% and 60%, v/v) were sequentially introduced into the tubes using a pipette. Centrifugation was carried out at 3000 RPM $min^{-1}$ for 3 minutes at 4°C. The spores in each sample were recovered at the interface of the two sucrose solutions using a pipette and thoroughly rinsed with distilled water through a 50μm sieve. Finally, spores obtained after extraction were enumerated under a binocular magnifying glass.

## Scaling up mycorrhizal inoculum production on peanut shells at the pilot site

The pilot site was equipped with an agricultural greenhouse (16 m length; 9 m width) covered with a windbreak line to ensure good ventilation, photoprotection and avoid overheating. Crushed peanut shells were used as substrate. The shells were moistened with tap water and

sterilized by cooking for 30 minutes in metal barrels heated with firewood by monitoring the cooking and avoiding burning (Fig 1).

The sterilized material was then distributed into plastic containers (54 cm x 80 cm); each container received 12.5 kg of sterile crushed peanut shells. Five treatments were tested. These included the four AMF starters provided by LCM and one control non-inoculated treatment. Corn (*Zea mays* L.) served as the host plant, with seeds soaked in tap water for 2 hours before being directly sown at a rate of 20 seeds per container. In each container, a total of 200 g of microbial starter was deposited at a depth of 2–3 cm at the same time as corn sowing. Each treatment comprised sixteen repetitions.

The multiplied inoculum was collected at the end of the growth phase after 3 months of cultivation and 3 weeks off watering. Aerial parts of the corn were removed, and the inoculum was harvested from each container by mixing thoroughly cut corn roots with the peanut shell substrate. Subsequently, samples of 100 g from each repetition were collected and mixed for each treatment to create a random sample inoculum for quality control at the laboratory in Dakar. Analysis of the chemical composition of the peanut shell, both before and after inoculum production, was conducted at the Laboratory of Analytical Means in Dakar (http://www.imago.ird.fr/).

## Quality control of the inoculum produced on peanut shells at the pilot site

Quality control consisted in estimating the mycorrhization rate as previously described. The spore extraction protocol from the peanut shell was however adapted, incorporating a preliminary step to remove plant debris. This phase involved placing 100 g of inoculum in 2 l of tap water in a container, stirring the mixture, and pouring it through a 500 µm sieve. This debris removal process was repeated three times. Once largest debris discarded, the protocol continued with the Gerdemann and Nicolson [29] method, spooning the water content (6 l) through

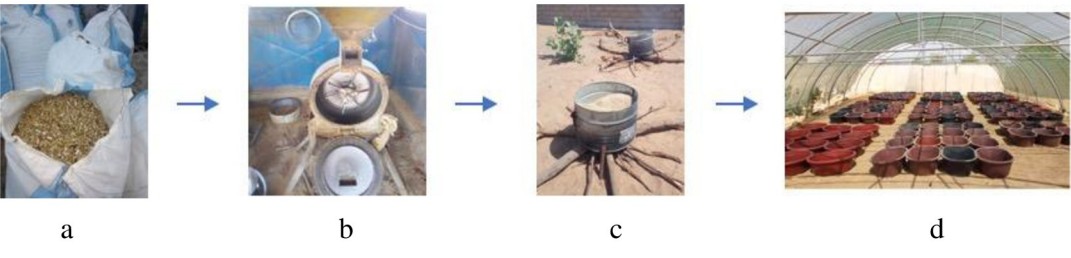

a b c d

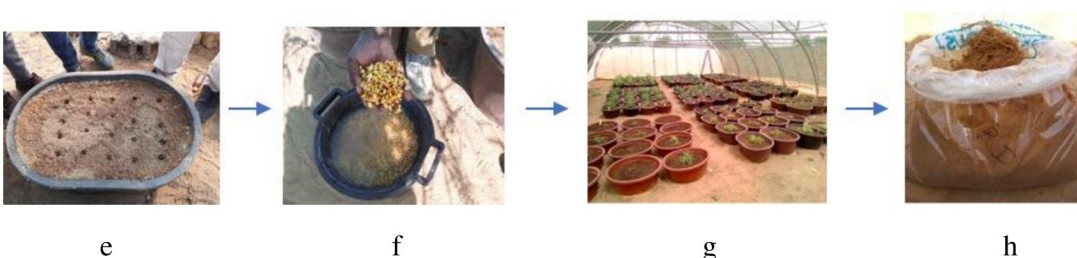

e f g h

**Fig 1. Plan and stapes of scaling-up of AMF inoculum production on pilot site.** a, not crushed peanut shells; b, shell grinding mill; c, sterilized material (metal barrels); d, plastic containers placed in greenhouse; e, sterile crushed peanut shells into plastic containers; f, soaked corn seeds; g, growing corn plants; h, collected multiplied inoculum (substrate and roots).

a series of sieves (400, 200, 100, and 50 μm). In each case, spores retained by the 200 μm and 50 μm sieves were collected in distilled water and transferred to centrifuge tubes. The subsequent steps of the protocol were then followed as described before. The spores obtained after extraction were counted under a binocular magnifying glass.

## Effect of salty water in watering plant and substrate sterilization on inoculum quality

To find out whether we could use water from the Darou Mousty site, which contains salt, and a non-sterile substrate, we assessed the impact of the salt content of the irrigation water and of substrate sterilization on the quality of the inoculum produced in a new experimental design under laboratory conditions. To guarantee infection and avoid possible infection specificity, starter comprising a mixture of the four strains in equal proportion, was used for this test. The inoculum was applied to sown corn on peanut shells under the following treatments in triplicates:

- Unsterilized substrate, inoculated (with the 4 AMF strains) and watered with unsalted water (LCM:NoSalty:No Sterile);

- Unsterilized substrate, inoculated and watered with salt water (LCM:Salty:No Sterile);

- Sterilized substrate, inoculated and watered with unsalted water (LCM:No Salty:Sterile);

- Sterilized substrate, inoculated and watered with salt water (LCM:Salty:Sterile);

- Sterilized support, not inoculated and watered with salt water (T:Salty:Sterile);

- Unsterilized support, uninoculated and watered with unsalted water (T:No Salty:No Sterile);

- Sterilized support, uninoculated and watered with non-salty water (T:No Salty:Sterile);

- Unsterilized support, uninoculated and watered with salt water (T:Salty:NoSterile).

In each case, the substrate consisting in sterilized or non-sterilized shells was distributed among 6 jars of 2 L each, and received ten grams of the AMF starter mixture as inoculum. Three pots were watered with drinking water, and three with a salted solution of 0.5% (w/v) of dietary salt (corresponding to salt concentration found in Darou Mousty irrigation water). Five corn seeds were sown into each pot.

## Statistical analyses

For the CMA production test on different agricultural residues in Leonard pots and jars, the effects of inoculation and support on the intensity of mycorrhization, the frequency of mycorrhization, and the number of spores were analyzed. Firstly, the normality of the data distribution for all variables was assessed using the Shapiro-Wilk test for each combination of factor levels. Subsequently, homogeneity of variance was evaluated. The effect of inoculation was assessed using a two-way analysis of variance (ANOVA) for the jar test and a three-way analysis of variance (ANOVA) for the Leonard jar test. To evaluate the salinity effect of salted water on inoculum quality, the same analysis as before (normality, homogeneity of variance) was conducted, and a three-way analysis of variance was employed to assess the effect. Statistical significance was considered at a level of $p < 0.05$. The Fisher method was used to separate the means at $P < 0.05$ level of significance. A Principal Component Analysis (PCA) was also constructed for this design. All statistical analyses were performed using R software (R Core Team, 2023) [34].

## Results

### Effect of agricultural residue supports on the production of AMF inoculum

The data analysis revealed a significant impact of corn inoculation on mycorrhization parameters, including intensity, frequency, and spore count, in agricultural residues, both within Leonard jars (Table 2) and pots (Table 3), as compared to the uninoculated control.

No statistically significant differences were observed among residues and upon addition of sand both under controlled conditions and under semi-controlled conditions (S1 and S2 Tables). However, the most favorable characteristics of the inoculum were consistently achieved on sand, peanut shell alone, or a combination of peanut shell and sand both under controlled or semi-controlled conditions (respectively Tables 2 and 3).

Peanut shell and sand exhibited superior performances for most mycorrhization parameters assessed: mycorrhization intensity, mycorrhization rate, and spore count, both in Leonard Jars and pots. Consequently, peanut shell was chosen as sustainable substrate to test the scaling up of inoculum production into a delocalized pilot site in Darou Mousty.

### Scaling-up production of mycorrhizal fungi inoculum on peanut shell at the farm in the rural zone

The mycorrhization parameters of the mass-produced inoculum on the peanut shell are presented in Table 4.

**Table 2. Characteristics of the inoculum produced on agricultural residues alone or mixed with sand in Leonard jars.**

| Substrate[a] | Mean[b] mycorrhization intensity (%) | Mean mycorrhization frequencies (%) | Mean number of spores (100⁻¹g of Substrate) |
|---|---|---|---|
| peanut shell | 53.9 | 90 | 712 |
| rice husk | 29.7 | 62 | 543 |
| ears of millet | 16.9 | 44 | 271 |
| bagasse | 7.1 | 25 | 18 |
| sand | 54.4 | 64 | 681 |
| substrate/sand (50/50) | | | |
| peanut shell | 29.2 | 55 | 639 |
| rice husk | 30.5 | 78 | 537 |
| ears of millet | 15.8 | 51 | 101 |
| bagasse | 4.3 | 7 | 21 |

[a]Uninoculated control plants were grown on each substrate and showed no mycorrhizal development.

[b]The mean data was obtained on sample composed of equal inoculum aliquots of 6 repetitions.

**Table 3. Characteristics of the inoculum produced on agricultural residues in pots.**

| Substrate[a] | Mean[b] mycorrhization intensity (%) | Mean mycorrhization frequencies (%) | Mean number of spores (100⁻¹g of Substrate) |
|---|---|---|---|
| peanut shell | 46.3 | 86 | 527 |
| rice husk | 37.4 | 84 | 503 |
| ears of millet | 20.48 | 31 | 278 |
| bagasse | 6.2 | 19 | 27 |
| sand | 41.7 | 68 | 548 |

[a]Uninoculated control plants were grown on each substrate and showed no mycorrhizal development.

[b]The mean data was obtained on sample composed of equal inoculum aliquots of 4 repetitions.

**Table 4. Characteristics of the inoculum produced on the peanut shell at the pilot site.**

| Characteristics | | *G. agregatum* IR27 | *G. fasciculatum* | *F. mosseae* | *R. irregularis* | Control |
|---|---|---|---|---|---|---|
| Mean Number of spores (100$^{-1}$g substrate) | Average[a] | **13.4 a** | **15.4 a** | **13.6 a** | **21.2 a** | **6.2 a** |
| | Std | 5.5 | 13.48 | 1.34 | 15.06 | 7.12 |
| Mean Mycorrhization intensity (%) | Average[a] | **19.76 a** | **13.66 ab** | **8.66 bc** | **9.58 b** | **0.91 c** |
| | Std | 6.55 | 5.95 | 2.84 | 4.12 | 0.81 |
| Mean Mycorrhization frequency (%) | Average[a] | **88.93 a** | **70.4 a** | **62.07 ab** | **42.20 bc** | **15.0 c** |
| | Std | 5.36 | 16.79 | 18.23 | 17.21 | 20.55 |

[a]Values for each characteristic followed by the same letter are not significantly different at $P < 0.05$ (Fisher test).

The statistical analysis revealed that the number of spores produced by mycorrhizal fungi did not exhibit significant differences between the strains, nor when compared to the control. However, the mycorrhization intensity demonstrated a highly significant difference (Pr = 8.07e-05), with the strains *Glomus aggregatum*, *Glomus fasciculatum*, and *Rhizophagus irregularis* (in descending order) displaying superior performance compared to the control (S3 Table). Notably, no significant distinction was observed between the *Glomus aggregatum* and *Glomus fasciculatum* strains regarding this parameter. Regarding mycorrhization frequencies, the strains *Glomus aggregatum*, *Glomus fasciculatum*, and *Funneliformis mosseae* exhibited significantly enhanced performance (in descending order) compared to the control (Pr = 3.28e-05).

However, when compared with the characteristics of the starters used for inoculum production (Table 1), such as the number of spores (1313 versus 13.4 for the *Glomus aggregatum* IR27 strain, for instance) and the intensity of mycorrhization (95% versus 8.66% for *Rhizophagus irregularis*), the infectious potential of the inoculum exhibited very low values (Table 4). Among the parameters potentially influencing inoculum quality, factors such as the chemical composition of peanut shell (particularly phosphorus content), quality of irrigation water, or the spore extraction method were investigated. Chemical analyses of the peanut shell composition before and after production did not reveal any significant differences (Table 5). Minor increases in carbon and phosphorus contents were noted, along with slight decreases in sodium and potassium contents, pH, and conductivity in the inoculum.

On the other hand, concerning the water utilized for irrigating the corn crop during inoculum production, the salt concentration was measured using a salinometer and registered a value of 5 g/l. According to a classification of salinity limits for irrigation water [35], this concentration corresponds to 0.78 dS/m and is considered as having an average salinity level. To evaluate the potential impact of water salinity on inoculum quality, another experiment was conducted in the laboratory as described in Materials and Methods section.

**Table 5. Chemical composition of peanut shell before and after the inoculum production.**

| Peanut shell[a] | total N (%) | total C (%) | total P (mg kg$^{-1}$) | Na (mg kg$^{-1}$) | K (mg kg$^{-1}$) | pH | Conductivity (µS/cm) |
|---|---|---|---|---|---|---|---|
| Before production | 0.79 | 38.98 | 575 | 2 566.0 | 1 632 | 6.04 | 0.42 |
| After production | 0.79 | 43.21 | 602.67 | 2 322.0 | 1 545.67 | 5.81 | 0.38 |

[a]The data are the averages of three aliquots for each peanut shell sample.

## Effect of irrigation water salinity and sterilization on the quality of mycorrhizal inoculum produced using peanut shells

The results of the 3-way ANOVA showed a significant interaction Substrate:Water:Treatments on the variables intensity and frequency of mycorrhization, and the interactions Substrate: Treatments and Water:Treatments on the variable number of spores (S2 Table).

The inoculum obtained on peanut shells from the LCM starter, watered with unsalted water, exhibited a superior mycorrhization profile (frequency: 33.33% and intensity rate: 7.33%). Notably, all samples watered with salty water displayed an absence of mycorrhization, except for the inoculum obtained from the LCM starter on the non-sterile substrate, attributed to the presence of exogenous strains. The sterile support yielded better results than the non-sterile one for frequency, intensity, and spore count (mean of 211 spores/100g of inoculum collected on sterile support compared to a mean of 79 spores/100g of inoculum collected on non-sterile support). The detrimental impact of saline irrigation water on propagule development was confirmed through mycorrhization parameters of control plants grown on non-sterile support in the presence of exogenous AMF fungi (Fig 2; S1 Fig).

Principal component analysis (PCA) carried out on the mycorrhization parameters (number of spores, mycorrhization frequency and intensity) in relation to the substrate (sterile and non-sterile) and to the irrigating water (salty and non-salty) showed that 99.5% of variance were explained by the first 2 principal components (PC1:93.7% and PC2:5.8%) (Fig 3; S2 Fig).

The analysis clearly separates on the first axis the treatment LCM:No_Salty:Sterile from other treatments (LCM:Salty:Sterile, LCM:No Salty:No_Sterile, LCM:Salty:No_Sterile, T:Salty: Sterile, T:No_Salty:No_Sterile, T:No_Salty:Sterile and T:Salty:No_Sterile) that appeared homogeneous on the first axis and which were also associated to lower mycorrhization capacity. The PCA results highlights that the LCM:No Salty:Sterile treatment stood out with a better mycorrhizal potential (spores, intensity and frequency) (Fig 3).

The results of the experiments are shown as box plots (Fig 2) and demonstrates that watering the corn with unsalted water improve all mycorrhization parameters. Water containing 0.5% salt inhibited the mycorrhization. This negative effect of salt was much more accentuated in unsterilized peanut shell substrate than in sterilized one (Fig 2).

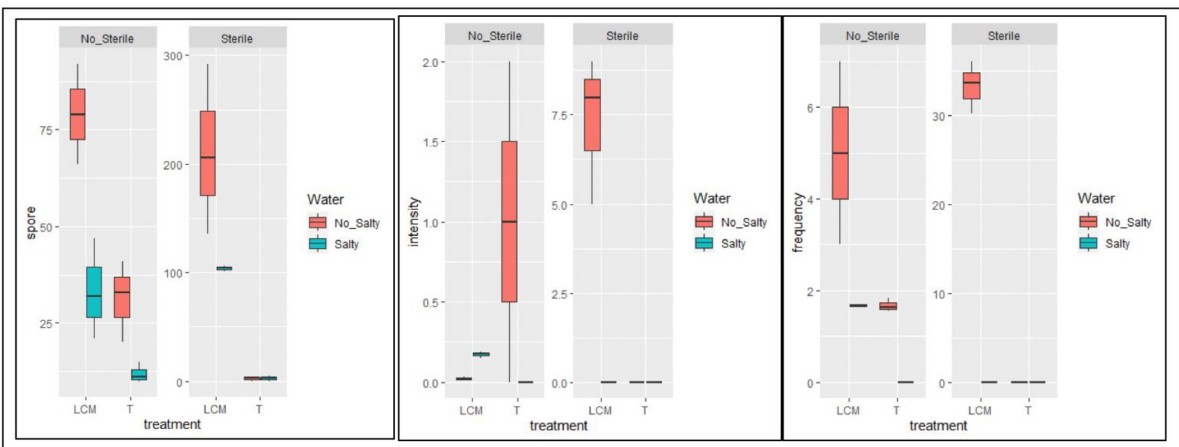

**Fig 2. Box plot showing the effect of using salt water and substrate sterilization on inoculum quality parameters (Spore number, mycorrhization intensity and frequency).** LCM refers to inoculation with the mix of LCM strains; T, uninoculated; Salty, irrigated with salted water (NaCl, 0.5% W/V); No Salty, irrigated with unsalted water; Sterile, sterilized peanut shell support; Non_Sterile, unsterilized peanut shell support.

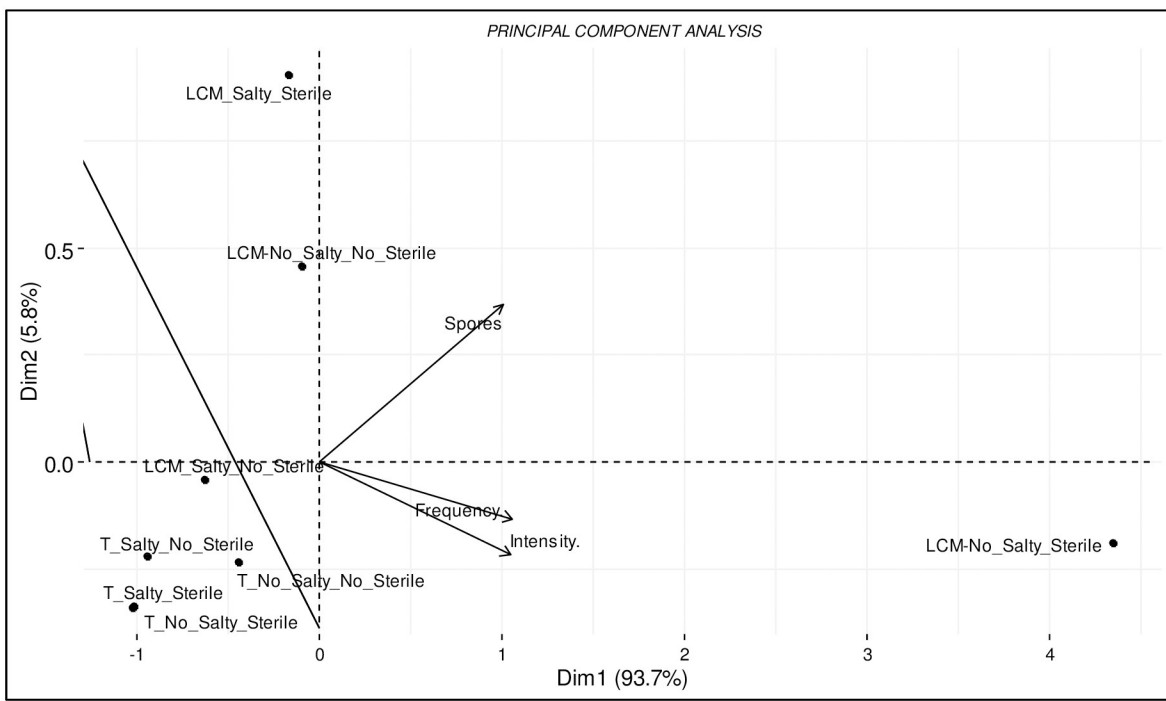

**Fig 3. Principal component analysis showing the effects of the treatments on the mycorhization using as variables the number of spores, mycorrhization intensity and frequency.** (:) means combination between treatments; LCM:No Salty:No Sterile, unsterilized support inoculated and watered with unsalted water; LCM:Salty:No Sterile, unsterilized support, inoculated and watered with salt water; LCM:No Salty:Sterile, sterilized support, inoculated and watered with unsalted water; LCM:SaltySterile, sterilized support, inoculated and watered with salt water; T:Salty:Sterile, sterilized support, not inoculated and watered with salt water; T:No_Salty:No Sterile, unsterilized support, uninoculated and watered with unsalted water; T:No_Salty:Sterile, sterilized support, not inoculated and watered with non-salty water and T:Salty:No Sterile, unsterilized support, uninoculated and watered with salt water.

## Improved method for extracting mycorrhizal spores from peanut shell substrate

Sterilization of the peanut shell by cooking for 30 minutes with firewood in the field led to acquiring a support with a significantly lower spore count compared to control plants (refer to Table 4 and S3 Table). Furthermore, an enhancement in the spore extraction method applied to the inoculum produced on peanut shells was achieved compared to the classic method on sand. The Fig 4 presents the results obtained using the two methods. The improved method resulted in collecting a significantly higher number of spores (F = 21.88; p<0.0001) both in the inoculum obtained in the lab from LCM starter (mean of 123.6 spores collected by the improved method vs. 106 by the classic method) and in the inoculum produced in Darou Mousty (81 spores collected by the improved method vs. 45 spores by the classic method).

## Discussion

Despite the central role of bio-inoculums in sustainable agriculture, the production and adoption of AMF in smallholder systems in Africa remain limited due to a lack of awareness and understanding. Research capacity and technological challenges, as identified by Mukhongo et al. [1], constitute significant barriers. The envisioned large-scale production method involves harnessing the natural environment while adhering to specific parameters critical for successful AMF inoculum development. By utilizing indigenous resources and optimizing the

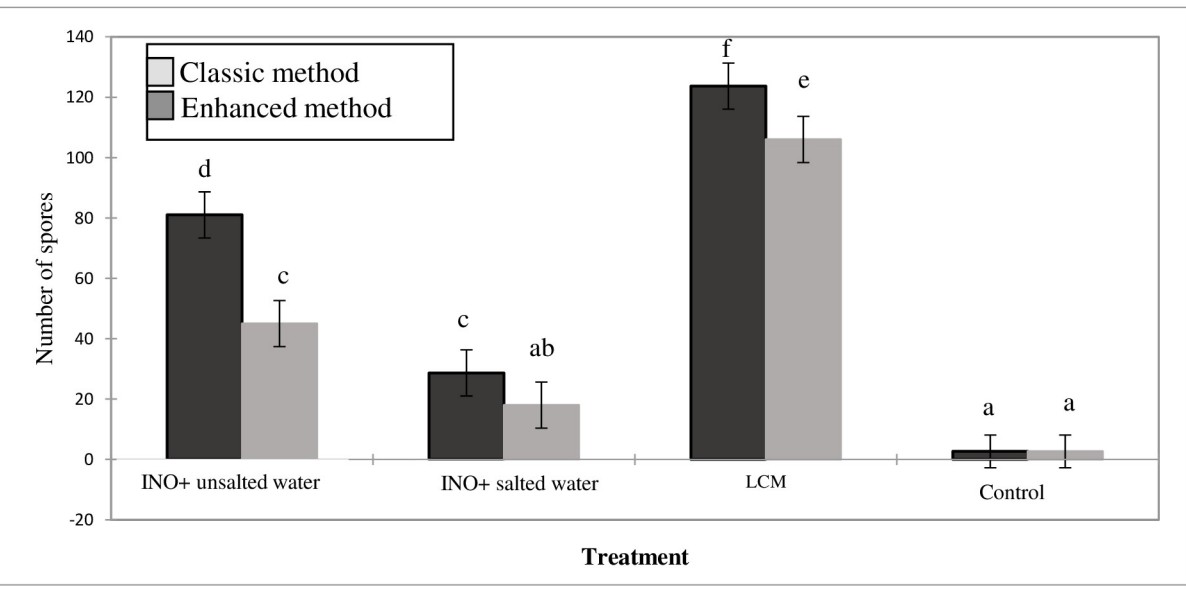

**Fig 4. Mean number of spores obtained by classical and improved extraction methods when using peanut shells as growth substrate.** The means of three repetitions per treatment followed by different letters differ according to the Fisher test (p = 0.05). Bars represent standard deviations from the mean (n = 3). The comparison is made within each treatment. INO, inoculum originated from the Darou Mousty pilot site obtained with salted and unsalted irrigation water; LCM, inoculum obtained from starter of Joint Microbiology Laboratory (LCM) on peanut shells in the laboratory; control, uninoculated corn grown in peanut shell.

available infrastructure, this approach seeks to overcome technological limitations and promote a cost-effective, environmentally friendly, and easily replicable process.

Basically, the inoculum production technique is simple. The process of AMF inoculum production typically involves monoculture starters or strain mixtures under controlled conditions in greenhouses, plant growth chambers, or occasionally in open-air farm settings [36–41]. Starters contain propagules in the form of spores, colonized root fragments, and hyphae [42, 43], and are multiplied on growing substrates such as peat, compost, vermiculite, sand, etc. [44]. These fungi, being obligate symbionts, necessitate host plant presence for cultivation. The most prevalent method for solid medium multiplication involves depositing colonized root fragments or AMF spores in a substrate sterilized at high temperatures and atmospheric pressure, or via gamma radiation. Subsequently, seeds of a plant capable of forming mycorrhizae are sown. After several months, the substrate and roots of the host plant can be used as inoculum [45].

Our study demonstrates the feasibility of transferring of this technology and know-how for the on-farm production of AMF inoculum from the laboratory to the field. The key components of this field-based production method include adaptation and control of relevant factors essential for AMF proliferation such as substrate composition and treatment, watering and cultivation processes. This requires adaptation to local climatic conditions and the selection of appropriate growing substrates, ensuring compatibility with the target AMF strains. The method emphasizes the use of low-cost, locally sourced materials that can create a conducive environment for AMF multiplication.

Several authors [1, 25] have enumerated key characteristics for an ideal inoculum support, encompassing good water retention capacity, optimal dry matter content, carbon, pH, porosity, ease of sterilization and adjustment, non-toxicity, and availability. The quality and effectiveness of the biostimulant during production and post-introduction into the soil are

influenced by temperature, humidity, substrate pH, and the host plant, thereby potentially impacting crop yields [29, 25, 46].

Agricultural residues, as opposed to the commonly used sand, emerge as a viable alternative for supporting inoculum production. Heavy, sand as a conventional support, poses challenges when transitioning to large-scale production, while numerous locally available and annually renewable organic materials from agricultural production stand as advantageous alternatives. In this study, peanut shell emerged as a promising substrate, undergoing partial decomposition during inoculum production due to corn growth, it may also subsequently contribute to soil amendment and restitution of organic matter (carbon, phosphorus, and nitrogen) upon field inoculation. Its use suggests potential accessibility of these elements to the inoculated cultures.

Studies by Saranya and Kumutha and Coelho et al. [47, 48] have emphasized the positive impact of adding organic matter to sand as a substrate for sporulation promotion, with optimal benefits dependent upon the specific organic source. The highest spore production was reported when vermicompost constituted 10% of the substrate [47] or when a mixture of sand, vermiculite, and 10% vermicompost was employed [48]. According to Saranya and Kumutha [47], compost has been identified as an excellent alternative substrate for mycorrhizal fungi multiplication, and biochar has also been recommended for large-scale production due to its influence on nutrient availability and organic matter stabilization. Further studies combining peanut shells enriched in compost or biochar that can be produced and acquired locally may improve AMF inoculum quality in the future.

Nevertheless, the selection of agricultural residues suitable for AMF production necessitates consideration of phosphorus content, given its influential role in regulating AMF propagule production [49]. Douds Jr [50] observed increased sweet potato yields in soils with high phosphorus content when using an inoculum containing a mixture of *Glomus intraradices* strains. Similarly, the addition of phosphorus to the irrigation solution of *P. miliaceum* plants promoted *G. etunicatum* sporulation, highlighting the potential adaptability of strains to high phosphorus conditions.

Among the studied strains, *Rhizophagus irregularis* exhibited significantly better sporulation under pilot site culture conditions, although not differing significantly from the control. In contrast, *Glomus agregatum* IR27 and *Glomus fasciculatum* showed significantly improved characteristics in terms of mycorrhization intensity (19.76% and 13.66%, respectively) and frequencies (88.93% and 70.4%, respectively), when compared to the control. Among commercially available products, R. *irregularis* and *F. mosseae* are the most used species, with R. *irregularis* prevailing in 39% of 68 products from 28 companies [51].

Salomon et al. [46] proposed a fundamental framework for quality control of commercial inocula, addressing essential criteria and control measures such as spore counts, root colonization, absence of pathogens eliminated through substrate sterilization, and support quality. A positive growth response in host plants and significant arbuscular mycorrhizal (AM) root colonization, with at least 20% of the root length colonized, are crucial indicators of inoculum viability. This experience validated the selection of a support meeting the criteria of availability, adaptability to plant and fungal development, and the ability to ensure high-quality inoculum. The results underscore the significance of substrate sterilization and the quality of irrigation water as critical parameters for obtaining superior-quality inoculum. Furthermore, the refinement of the spore extraction protocol from peanut shell, involving the incorporation of a 500 μm sieve for enhanced spore passage, resulted in improved inoculum characteristics. Numerous and lightweight plant debris, obstructing spore passage through the initial sieve, were effectively addressed, leading to a significant increase in spore quantity collected in subsequent sieves.

On-farm production of AM fungus inoculum appears to be a viable option in labor-intensive agricultural contexts typical of developing countries, using materials easily accessible to. This approach positions the on-farm production of local AMF inoculum adapted to the soil and climate as an interesting alternative to the importation of biofertilizers based on mycorrhizal fungi and opens the prospect of practical local application.

## Conclusion

A technical package for scaling up the production of Arbuscular Mycorrhizal Fungi inoculum was developed in the field. The production protocol under the tested conditions and the quality control protocol for the produced inoculum were optimized considering the production substrate based on agricultural residues and the salinity of watering. Based on this experience, the establishment of a local sector for the production and distribution of endomycorrhizal fungi inoculum in Senegal appears feasible. Validated in a delocalized site, this optimized process not only addresses the technical challenges and constraints of traditional methods but also emphasizes the adaptability of the process to local conditions. This innovative approach facilitates the availability of high-quality AMF inoculum for smallholding farmers, offering a sustainable and cost-effective solution to enhance crop productivity, reduce reliance on external inputs, and promote environmentally friendly agricultural practices.

## Supporting information

**S1 Table. Characteristics of the inoculum produced on agricultural residues alone or mixed with sand in Leonard jars.** [a]The mean data was obtained on sample composed of equal inoculum aliquots of 6 repetitions. Arbuscular mycorrhizal fungi root colonization rate was assessed using the method outlined by Trouvelot et al. [36].
(DOCX)

**S2 Table. Characteristics of the inoculum produced on agricultural residues in pots.** [a]The mean data was obtained on sample composed of equal inoculum aliquots of 4 repetitions. Arbuscular mycorrhizal fungi root colonization rate was assessed using the method outlined by Trouvelot et al. [36].
(DOCX)

**S3 Table. Characteristics of the inoculum produced on the peanut shell at the pilot site (ANOVA with R, 2023).**
(DOCX)

**S1 Fig. Effects of the treatments on the mycorhization using as variables the number of spores, mycorrhization intensity and frequency (Box plot with R. 2023).**
(DOCX)

**S2 Fig. Type III sum squares analysis of the treatment effects on the mycorhization using as variables spore number.** mycorrhization intensity and frequency obtained on corn plants (ANOVA with R. 2023).
(DOCX)

**S3 Fig. Mean number of spores obtained by classical and improved extraction methods when using peanut shells as growth substrate (ANOVA with XLSTAT).**
(DOCX)

## Acknowledgments

The authors thank the colleagues and direction of the Joint Microbiology Laboratory IRD-ISRA-UCAD (LCM), the Senegalese Innovative Business Incubator (InnoDev/UCAD) also for supporting and the authorities of the commune of Darou Mousty and the farmers for the facilities granted to this work.

## Author Contributions

**Conceptualization:** Tatiana Krasova Wade.

**Data curation:** Christian Valentin Nadieline.

**Formal analysis:** Christian Valentin Nadieline, Cheikh Ndiaye, Oumar Sadio.

**Funding acquisition:** Yoro Idrissa Thioye, Cheikh Mouhamed Fadel Kébé, Tatiana Krasova Wade.

**Investigation:** Christian Valentin Nadieline, Antoine le Quéré, Cheikh Ndiaye.

**Methodology:** Christian Valentin Nadieline, Antoine le Quéré, Cheikh Ndiaye, Amadou Abib Diène, Francis Do Rego.

**Project administration:** Tatiana Krasova Wade.

**Resources:** Amadou Abib Diène, Francis Do Rego, Cheikh Mouhamed Fadel Kébé.

**Supervision:** Tatiana Krasova Wade.

**Validation:** Antoine le Quéré, Tatiana Krasova Wade.

**Writing – original draft:** Christian Valentin Nadieline, Tatiana Krasova Wade.

**Writing – review & editing:** Antoine le Quéré, Marc Neyra.

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
