## [Decision Letter · Decision Letter 0]

21 May 2024

PONE-D-24-13955Development of on-farm AMF inoculum production for sustainable agriculture in SenegalPLOS ONE

Dear Dr. Krasova Wade,

Thank you for submitting your manuscript to PLOS ONE. After careful consideration, we feel that it has merit but does not fully meet PLOS ONE’s publication criteria as it currently stands. Therefore, we invite you to submit a revised version of the manuscript that addresses the points raised during the review process.

We look forward to receiving your revised manuscript.

Kind regards,

Sofia Isabel Almeida Pereira

Academic Editor

PLOS ONE

Journal Requirements:

 [OAPI funding 27714 and IRD funding MySen].  

[This work was supported by the African Intellectual Property Organization (OAPI) funding 27714 and the Senegalese Innovative Business Incubator (INNODEV) between 2013 and 2016, and by the French National Research Institute for Sustainable Development (IRD) funding MySen between 2017 and 2019. The authors thank the management of the Joint Microbiology Laboratory IRD-ISRA-UCAD (LCM) for supporting and the authorities of the commune of Darou Mousty and the farmers for the facilities granted to this work.]

 [OAPI funding 27714 and IRD funding MySen]

5. We note that your Data Availability Statement is currently as follows: [All relevant data are within the manuscript and its Supporting Information files]

Reviewers' comments:

Reviewer's Responses to Questions

**Comments to the Author**

1. Is the manuscript technically sound, and do the data support the conclusions?

Reviewer #1: Partly

Reviewer #2: Yes

Reviewer #3: Yes

2. Has the statistical analysis been performed appropriately and rigorously? 

Reviewer #1: I Don't Know

Reviewer #2: I Don't Know

Reviewer #3: Yes

3. Have the authors made all data underlying the findings in their manuscript fully available?

Reviewer #1: Yes

Reviewer #2: Yes

Reviewer #3: Yes

4. Is the manuscript presented in an intelligible fashion and written in standard English?

Reviewer #1: No

Reviewer #2: Yes

Reviewer #3: Yes

5. Review Comments to the Author

Reviewer #1: Thank you for the opportunity to review for the esteemed journal, PLOS ONE.

Regarding the manuscript, I regret to say that it may not align with the interests of your journal's readership. Major revisions are necessary before considering it for publication in the PLOS ONE. Nonetheless, I recommend the authors to undertake substantial revisions to improve the manuscript.

It is advisable to include some information about materials and methods in the abstract.

The referencing style should adhere to the journal's guidelines.

The abstract is not presented in a manner that the readers understand the main results of this study.

The Materials and Methods section requires additional details about experimental conditions.

There is a lack of comprehensive description regarding experimental design, materials, and methods.

Please explicitly state whether the objectives are novel.

The below updated references would be useful:

https://doi.org/10.1016/B978-0-443-15884-1.00002-6

https://doi.org/10.1016/j.indcrop.2022.115557

https://doi.org/10.1016/j.eja.2021.126283

https://doi.org/10.1111/jam.15326

https://doi.org/10.1016/j.indcrop.2016.06.024

https://doi.org/10.1016/j.eja.2016.01.006

The introduction lacks coherence, with some sentences appearing conceptually disconnected, particularly towards the conclusion. Given the significance of the study, more attention should be devoted to refining the introduction.

Clear articulation of hypotheses and objectives is essential.

The Materials and Methods section necessitates restructuring for clarity, as some paragraphs are poorly organized, leading to confusion.

Insufficient presentation, especially in terms of mean comparison, and inadequate description of results are evident. Additionally, there is an overabundance of poorly annotated illustrations and tables.

It is challenging to discern how conclusions were derived from the illustrations in some instances, while other effects were not adequately presented.

The discussion lacks depth, primarily consisting of a list of references for each measured variable. There is a notable absence of analysis on experimental processes and practical implications. The discussion fails to contextualize the experimental findings or justify the necessity of the study compared to existing literature.

The discussion lacks integration among different variables measured in the experiment, failing to draw simultaneous conclusions from their measurements.

Reviewer #2: Abstract

- Lion 41-42: I think this information is inaccurate. There is a lot of research conducted in African countries in this regard, and commercial biofertilizer compounds contain mycorrhizal fungi for use in agricultural fields.

Introduction

- Although the introduction section is large, it does not contain specific information about the study factors

- Most of the introduction is general talk that needs to be shortened

- The introduction needs to be strengthened with previous studies that address the study factors

Reviewer #3: Manuscript ID: PONE-D-24-13955

Development of on-farm AMF inoculum production for sustainable agriculture in Senegal

Dear editor,

The paper is original. The topic is interesting and may have application potential. The authors aim to investigate the feasibility of producing mycorrhizal inoculum on a semi-industrial farm scale. Agricultural residues were tested as replacements of sand. In the other experiments they have tested the effects of salinity water and soil sterilization on AMF prolifiration and production. The experiments are well designed and the data are more or less enough to interpret the results. The text is written relatively well. The results were relatively well-presented and discussed but I have some more concerns as they have been mentioned in the pdf file as note which has been attched.

6. PLOS authors have the option to publish the peer review history of their article (what does this mean?). If published, this will include your full peer review and any attached files.

Reviewer #1: No

Reviewer #2: No

Reviewer #3: No

---

## [Author Response · Author response to Decision Letter 0]

2 Jul 2024

Dear Editor,

The Authors would like to thank the editorial team as well as reviewers for their careful and

constructive comments. We have been through Journal Requirements and Reviewers concerns and

made revisions accordingly. The revised manuscript submitted integrates all changes made and we

have responded to each point below using green font to facilitate the reading.

The PLOS ONE style templates can be found at

https://journals.plos.org/plosone/s/file?id=ba62/PLOSOne_formatting_sample_title_authors_affiliations.pd

f.

Response to Editor point 1:

The manuscript has been reviewed and formatted following PLOS ONE’s requirements.

2. We note that the grant information you provided in the ‘Funding Information’ and ‘Financial Disclosure’

sections do not match.

When you resubmit, please ensure that you provide the correct grant numbers for the awards you received

for your study in the ‘Funding Information’ section.

[OAPI funding 27714 and IRD funding MySen].

Please state what role the funders took in the study. If the funders had no role, please state: ""The funders

had no role in study design, data collection and analysis, decision to publish, or preparation of the

manuscript.""

Please include this amended Role of Funder statement in your cover letter; we will change the online

submission form on your behalf.

[This work was supported by the African Intellectual Property Organization (OAPI) funding 27714 and the

Senegalese Innovative Business Incubator (INNODEV) between 2013 and 2016, and by the French

National Research Institute for Sustainable Development (IRD) funding MySen between 2017 and 2019.

The authors thank the management of the Joint Microbiology Laboratory IRD-ISRA-UCAD (LCM) for

supporting and the authorities of the commune of Darou Mousty and the farmers for the facilities granted to

this work.]

We note that you have provided funding information that is not currently declared in your Funding

Statement. However, funding information should not appear in the Acknowledgments section or other areas

of your manuscript. We will only publish funding information present in the Funding Statement section of

the online submission form.

Please remove any funding-related text from the manuscript and let us know how you would like to update

your Funding Statement. Currently, your Funding Statement reads as follows:

[OAPI funding 27714 and IRD funding MySen]

Please include your amended statements within your cover letter; we will change the online submission

form on your behalf.

Thank for these clarifications, regarding points 2, 3 and 4, we have now removed any funding related

text from the manuscript and sections “Funding information” and “Financial Disclosure” have been

corrected and submitted online as follow:

Funding information: “This work was supported by the African Intellectual Property Organization

(OAPI) funding 27714 received in 2014 and by the French National Research Institute for Sustainable

Development (IRD) funding Coup de pouce “MySen” received in 2017.

Financial Disclosure: “The funders had no role in study design, data collection and analysis, decision to

publish, or preparation of the manuscript.”

5. We note that your Data Availability Statement is currently as follows: [All relevant data are within the

manuscript and its Supporting Information files]

Please confirm at this time whether or not your submission contains all raw data required to replicate the

results of your study. Authors must share the “minimal data set” for their submission. PLOS defines the

minimal data set to consist of the data required to replicate all study findings reported in the article, as well

as related metadata and methods (https://journals.plos.org/plosone/s/data-availability#loc-minimal-data-set-

definition).

Authors do not need to submit their entire data set if only a portion of the data was used in the reported

study.

If your submission does not contain these data, please either upload them as Supporting Information files or

deposit them to a stable, public repository and provide us with the relevant URLs, DOIs, or accession

numbers. For a list of recommended repositories, please

see https://journals.plos.org/plosone/s/recommended-repositories.

If there are ethical or legal restrictions on sharing a de-identified data set, please explain them in detail

(e.g., data contain potentially sensitive information, data are owned by a third-party organization, etc.) and

who has imposed them (e.g., an ethics committee). Please also provide contact information for a data

access committee, ethics committee, or other institutional body to which data requests may be sent. If data

are owned by a third party, please indicate how others may request data access.

Response to Editor point 5: please accept our apologies for not including these data in the first

version of the manuscript. We have now included as Supporting information all raw data obtained as

new dataset files for each one of the tables and figures presented in this work.

Supporting information related to raw data linked to the submitted work:

S1 dataset: raw data refereeing to Table 1 “Mycorrhization parameters (mean data) of AMF starters

used for inoculum production on the pilot site”

S2 dataset: raw data refereeing to Table 2 “Characteristics of the inoculum produced on agricultural

residues alone or mixed with sand in Leonard jars”

S3 dataset: raw data refereeing to Table 3 “Characteristics of the inoculum produced on agricultural

residues in pots”

S4 dataset: raw data refereeing to Table 4 “Characteristics of the inoculum produced on the peanut

shell at the pilot site”

S2 and S3 Fig dataset: raw data refereeing to Fig 2 “Box plot showing the effect of using salt water

and substrate sterilization on inoculum quality“ and Fig 3 “Principal Component Analysis showing

the effects of the treatments on the mycorhization using as variables the number of spores,

mycorrhization intensity and frequency“

.

S4 Fig dataset: raw data refereeing to Fig 4 “Mean number of spores obtained by classical and

improved extraction methods when using peanut shells as growth substrate“

.

6. We note that you have included the phrase “data not shown” in your manuscript. Unfortunately, this does

not meet our data sharing requirements. PLOS does not permit references to inaccessible data. We require

that authors provide all relevant data within the paper, Supporting Information files, or in an acceptable,

public repository. Please add a citation to support this phrase or upload the data that corresponds with these

findings to a stable repository (such as Figshare or Dryad) and provide and URLs, DOIs, or accession

numbers that may be used to access these data. Or, if the data are not a core part of the research being

presented in your study, we ask that you remove the phrase that refers to these data.

Response to Editor point 6: we have removed this part of the sentence as it is not dispensable and

was not appropriate as sand sterilization was not used in the present study.

The new sentence is now:” The shells were moistened with tap water and sterilized by cooking for 30

minutes in metal barrels heated with firewood by monitoring the cooking and avoiding burning (S1

Fig.).

Review Comments to the Author

Please use the space provided to explain your answers to the questions above. You may also include

additional comments for the author, including concerns about dual publication, research ethics, or

publication ethics. (Please upload your review as an attachment if it exceeds 20,000 characters)

Reviewer #1: Thank you for the opportunity to review for the esteemed journal, PLOS ONE.

Regarding the manuscript, I regret to say that it may not align with the interests of your journal's

readership. Major revisions are necessary before considering it for publication in the PLOS ONE.

Nonetheless, I recommend the authors to undertake substantial revisions to improve the manuscript.

We thank Reviewer 1 for his recommendations. We believe that the proposed study involving

multiple actors is of wide interest and fits to PLOS ONE aims. We have modified the text so as to

clarify all sections as recommended by reviewer 1, please see below for point by point responses to

reviewer 1 comments and associated revisions made in the main text.

It is advisable to include some information about materials and methods in the abstract.

> Additional information about materials and methods has been included in the abstract.

The referencing style should adhere to the journal's guidelines.

>The referencing style has been revised.

The abstract is not presented in a manner that the readers understand the main results of this study.

> We have added the following text in the Abstract section regarding materials and methods: “Peanut

shell, rice husk, sugar cane bagasse and millet ears were tested in Leonard jars and pots as alternatives to

conventional sand production substrate for the multiplication of mycorrhizal fungi Glomus aggregatum

IR27, Funneliformis mosseae, Rhizophagus irregulares and Glomus fasciculatum R10.”

and main results of the study: “Significant results were obtained on the peanut shell. Under mass

production conditions in farm scale, Glomus aggregatum IR27 showed the best mycorization

characteristics with 19.76% intensity and 88.93% frequencies. The study highlighted the critical

considerations of irrigation water salt content and substrate sterilization as essential parameters to ensure

optimal development of mycorrhizal propagules. Water containing 0.5% salt inhibited the mycorrhization.

This negative effect of salt was much more accentuated in unsterilized peanut shell substrate than in

sterilized one”

The Materials and Methods section requires additional details about experimental conditions.

There is a lack of comprehensive description regarding experimental design, materials, and methods.

> Details about materials and methods for the production of in farm AMF production have been

described as a new figure as supporting information (please see Fig 1).

Please explicitly state whether the objectives are novel.

> We have emphasized the novelty of the objectives to produce on farm AMF by producers using

“crop residues…” in the Introduction: “This experimental enterprise constitutes a pioneering initiative,

potentially replicable in other agricultural areas. Its sustainability relies on the integration of renewable

agricultural residues into the mycorrhizal inoculum production path in order to ensure the availability of the

product on a large scale.”

The below updated references would be useful:

https://doi.org/10.1016/B978-0-443-15884-1.00002-6

https://doi.org/10.1016/j.indcrop.2022.115557

https://doi.org/10.1016/j.eja.2021.126283

https://doi.org/10.1111/jam.15326

https://doi.org/10.1016/j.indcrop.2016.06.024

https://doi.org/10.1016/j.eja.2016.01.006

> We have decided to include three references (Seyyed Ali Sadegh Sadat Darakeh et al., 2022,

Weria Weisany, 2021 and Loftabadi et al., 2022) that we think will improve the manuscript; we

do not think that the other proposed references are essential for the comprehension of the article.

The introduction lacks coherence, with some sentences appearing conceptually disconnected, particularly

towards the conclusion. Given the significance of the study, more attention should be devoted to refining

the introduction.

Clear articulation of hypotheses and objectives is essential.

> We have revised the introduction, the conclusion connects well with the objectives now.

“This study endeavors to explore the feasibility of producing elite AMF inoculum on a semi-industrial

scale within agricultural production sites. The simplicity and efficiency of the technology lays the

foundations for the creation of a local sector for the production and distribution of inoculum of

endomycorrhizal fungi in Senegal. Ultimately, it contributes to stimulate the integration of symbiotic

microorganisms into agricultural practices in the form of bio-stimulants.”

The Materials and Methods section necessitates restructuring for clarity, as some paragraphs are poorly

organized, leading to confusion.

> Details about materials and methods for the production of in farm AMF production have been

described as a new figure as supporting information (please see S1 Fig).

Insufficient presentation, especially in terms of mean comparison, and inadequate description of results are

evident. Additionally, there is an overabundance of poorly annotated illustrations and tables.

> The tables have been transferred to the supporting information and the details have been provided

in terms of mean comparison. Additional data was provided in the supporting information.

It is challenging to discern how conclusions were derived from the illustrations in some instances, while

other effects were not adequately presented.

>The conclusion section has been revised and more focused on the present study. The prospects for

increasing of AMF production units have been identified.

The discussion lacks depth, primarily consisting of a list of references for each measured variable. There is

a notable absence of analysis on experimental processes and practical implications. The discussion fails to

contextualize the experimental findings or justify the necessity of the study compared to existing literature.

>The Discussion section has been revised and was more contextualized. The effort was made to

reconcile the results obtained with data from the literature. The connection was made with the

objectives.

The discussion lacks integration among different variables measured in the experiment, failing to draw

simultaneous conclusions from their measurements.

> We have taken these suggestions into account and revised all discussion section.

Reviewer #2:

Abstract

- Lion 41-42: I think this information is inaccurate. There is a lot of research conducted in African

countries in this regard, and commercial biofertilizer compounds contain mycorrhizal fungi for use in

agricultural fields.

> We are observed that industrial inoculum production in Africa is limited to a few countries.

Commercial products do not often contain native strains adapted to all environments. They are not

accessible to farmers. This study aims to develop a simple protocol for producing quality inoculum

whose know-how can be easily transferred to farmers to ensure their autonomy.

Introduction

- Although the introduction section is large, it does not contain specific information about the study factors

- Most of the introduction is general talk that needs to be shortened

- The introduction needs to be strengthened with previous studies that address the study factors

> The Introduction section has been improved: shortened, enriched with bibliographic data and

focused on the objectives and originality:

“This experimental enterprise constitutes a pioneering initiative, potentially replicable in other agricultural

areas. Its sustainability relies on the integration of renewable agricultural residues into the mycorrhizal

inoculum production path in order to ensure the availability of the product on a large scale. This study

endeavors to explore the feasibility of producing elite AMF inoculum on a semi-industrial scale within

agricultural production sites. The simplicity and efficiency of the technology lays the foundations for the

creation of a local sector for the production and distribution of inoculum of endomycorrhizal fungi in

Senegal. 

---

## [Decision Letter · Decision Letter 1]

24 Jul 2024

PONE-D-24-13955R1Development of on-farm AMF inoculum production for sustainable agriculture in SenegalPLOS ONE

Dear Dr. Krasova Wade,

Thank you for submitting your manuscript to PLOS ONE. After careful consideration, we feel that it has merit but does not fully meet PLOS ONE’s publication criteria as it currently stands. Therefore, we invite you to submit a revised version of the manuscript that addresses the points raised during the review process.

We look forward to receiving your revised manuscript.

Kind regards,

Sofia Isabel Almeida Pereira

Academic Editor

PLOS ONE

Journal Requirements:

Reviewers' comments:

Reviewer's Responses to Questions

**Comments to the Author**

1. If the authors have adequately addressed your comments raised in a previous round of review and you feel that this manuscript is now acceptable for publication, you may indicate that here to bypass the “Comments to the Author” section, enter your conflict of interest statement in the “Confidential to Editor” section, and submit your "Accept" recommendation.

Reviewer #1: (No Response)

Reviewer #2: All comments have been addressed

2. Is the manuscript technically sound, and do the data support the conclusions?

Reviewer #1: (No Response)

Reviewer #2: (No Response)

3. Has the statistical analysis been performed appropriately and rigorously? 

Reviewer #1: (No Response)

Reviewer #2: (No Response)

4. Have the authors made all data underlying the findings in their manuscript fully available?

Reviewer #1: (No Response)

Reviewer #2: (No Response)

5. Is the manuscript presented in an intelligible fashion and written in standard English?

Reviewer #1: (No Response)

Reviewer #2: (No Response)

6. Review Comments to the Author

Reviewer #1: The authors revised the entire manuscript according to the recommendations of the reviewers and the editor. It has been significantly improved, and the discussion is now sufficiently critical and can therefore be considered for acceptance

Reviewer #2: The introduction has been greatly shortened. Please add some research to the introduction that highlight the importance of mycorrhiza on plant production, especially in arid and semi-arid regions such as Africa. For example: https://doi.org/10.1016/j.rhisph.2024.100852

7. PLOS authors have the option to publish the peer review history of their article (what does this mean?). If published, this will include your full peer review and any attached files.

Reviewer #1: No

Reviewer #2: No

---

## [Author Response · Author response to Decision Letter 1]

7 Aug 2024

Dear Editor,

The Authors would like to thank the editorial team as well as reviewers for their constructive comments. We have made revisions accordingly the reviewers’ comments. The revised manuscript submitted integrates all changes made and we have responded to each point below using green font to facilitate the reading.

Journal Requirements:

>The reference list has been revised. No reference has been retracted. Four new references were cited as below.

Review Comments to the Author

6. Review Comments to the Author

Reviewer #1: The authors revised the entire manuscript according to the recommendations of the reviewers and the editor. It has been significantly improved, and the discussion is now sufficiently critical and can therefore be considered for acceptance.

>We thank Reviewer 1 for his positive appreciations. 

Reviewer #2: The introduction has been greatly shortened. Please add some research to the introduction that highlight the importance of mycorrhiza on plant production, especially in arid and semi-arid regions such as Africa. For example: https://doi.org/10.1016/j.rhisph.2024.100852

>We thank Reviewer 2 for his positive appreciations. Additional researches have been mentioned in the introduction that we think will improve the manuscript; 

“Additionally, it was demonstrated across diverse agro-ecological zones adaptability to saline and water stress conditions [19] and improving of plant production in arid and semi-arid zones [20, 21, 22, 23]”

The references’ list has been updated:

20. Shalaby OA, Ramadan MEl-SR. Mycorrhizal colonization and calcium spraying modulate physiological and antioxidant responses to improve pepper growth and yield under salinity stress. Rhizosphere. 2024;29,100852. https://doi.org/10.1016/j.rhisph.2024.100852.

21. Ould Amer S, Aliat T, Kucher DE, Bensaci OA, Rebouh NY. Investigating the potential of arbuscular mycorrhizal fungi in mitigating water deficit effects on durum wheat (Triticum durum Desf.). Agriculture. 2023;13(3):552-68. https://doi.org/10.3390/agriculture13030552.

22. Duan H-X, Luo C-L, Li J-Y, Wang B-Z, Naseer M, Xiong Y-C. Improvement of wheat productivity and soil quality by arbuscular mycorrhizal fungi is density- and moisture-dependent. Agron Sustain Develop. 2021;41:3-12. https://doi.org/10.1007/s13593-020-00659-8.

23. Wahab A, Muhammad M, Munir A, Abdi G, Zaman W, Ayaz A et al. Role of arbuscular mycorrhizal fungi in regulating growth, enhancing productivity, and potentially influencing ecosystems under abiotic and biotic Stresses. Plants. 2023;12:3102-40. https://doi.org/10.3390/plants12173102.

---

## [Editor Report · Decision Letter 2]

21 Aug 2024

Development of on-farm AMF inoculum production for sustainable agriculture in Senegal

PONE-D-24-13955R2

Dear Dr. Wade,

We’re pleased to inform you that your manuscript has been judged scientifically suitable for publication and will be formally accepted for publication once it meets all outstanding technical requirements.

Kind regards,

Sofia Isabel Almeida Pereira

Academic Editor

PLOS ONE

---

## [Editor Report · Acceptance letter]

4 Sep 2024

PONE-D-24-13955R2 

PLOS ONE

Dear Dr. Krasova Wade, 

I'm pleased to inform you that your manuscript has been deemed suitable for publication in PLOS ONE. Congratulations! Your manuscript is now being handed over to our production team.

Kind regards, 

on behalf of

Dr. Sofia Isabel Almeida Pereira 

Academic Editor

PLOS ONE